# Epithelial-Myeloid cell crosstalk regulates acinar cell plasticity and pancreatic remodeling in mice

Yaqing Zhang[1], Wei Yan[1,2], Esha Mathew[3], Kevin T Kane[1], Arthur Brannon III[3,4], Maeva Adoumie[5], Alekya Vinta[5], Howard C Crawford[6,7,8], Marina Pasca di Magliano[1,3,8,9]*

[1]Department of Surgery, University of Michigan, Ann Arbor, United States; [2]Department of Pathology, Xijing Hospital, Fourth Military Medical University, Xi'an, China; [3]Program in Cellular and Molecular Biology, University of Michigan, Ann Arbor, United States; [4]Medical Scientist Training Program, University of Michigan, Ann Arbor, United States; [5]College of Literature, Science, and the Arts, University of Michigan, Ann Arbor, United States; [6]Department of Molecular and Integrative Physiology, University of Michigan, Ann Arbor, United States; [7]Department of Internal Medicine, University of Michigan, Ann Arbor, United States; [8]Comprehensive Cancer Center, University of Michigan, Ann Arbor, United States; [9]Department of Cell and Developmental Biology, University of Michigan, Ann Arbor, United States

*For correspondence:
marinapa@umich.edu

Competing interests: The authors declare that no competing interests exist.

**Abstract** Dedifferentiation of acini to duct-like cells occurs during the physiologic damage response in the pancreas, but this process can be co-opted by oncogenic Kras to drive carcinogenesis. Myeloid cells infiltrate the pancreas during the onset of pancreatic cancer, and promote carcinogenesis. Here, we show that the function of infiltrating myeloid cells is regulated by oncogenic Kras expressed in epithelial cells. In the presence of oncogenic Kras, myeloid cells promote acinar dedifferentiation and carcinogenesis. Upon inactivation of oncogenic Kras, myeloid cells promote re-differentiation of acinar cells, remodeling of the fibrotic stroma and tissue repair. Intriguingly, both aspects of myeloid cell activity depend, at least in part, on activation of EGFR/MAPK signaling, with different subsets of ligands and receptors in different target cells promoting carcinogenesis or repair, respectively. Thus, the cross-talk between epithelial cells and infiltrating myeloid cells determines the balance between tissue repair and carcinogenesis in the pancreas.
DOI: https://doi.org/10.7554/eLife.27388.001

## Introduction

The highly specialized epithelial cells in the adult pancreas (*Slack, 1995*) derive from common progenitors during embryogenesis (*Cano et al., 2007*; *Gittes, 2009*; *Means and Leach, 2001*). The cells forming the exocrine pancreas, namely acinar, ductal and centroacinar cells, are believed to constitute the likely origin of pancreatic ductal adenocarcinoma, the most common type of pancreatic cancer and one of the deadliest human malignancies (for review, see (*Puri and Hebrok, 2010*; *Rooman and Real, 2012*; *Stanger and Dor, 2006*; *Zorn and Wells, 2007*)). Kras mutations are common in human pancreatic cancer (*Bailey et al., 2016*; *Hezel et al., 2006*; *Jones et al., 2008*), and are present with high frequency in the precursor lesions to pancreatic cancer known as Pancreatic Intraepithelial Neoplasia (PanIN) (*Kanda et al., 2012*; *Klimstra and Longnecker, 1994*). Genetically engineered mice that express oncogenic Kras in the pancreas develop PanINs that progress to

**eLife digest** The pancreas contains many types of highly specialized cells. For example, the acinar cells produce enzymes that help to digest food, and the ductal cells build the ducts to transport these enzymes to the gut. When the pancreas gets injured, the acinar cells start to transform into duct-like cells. The cells can revert to normal acinar cells once the tissue has repaired itself.

However, when a protein named Kras becomes faulty, the transformed acinar cells can no longer revert to normal ones. Scientists believe that is one of the first signs of pancreatic cancer, as mutated Kras proteins are very common in this disease. Injury and cancer both attract immune cells to the pancreas, including a type called myeloid cells. However, until now it was not known how myeloid cells and acinar cells interact.

In 2012, scientist showed that when the faulty Kras protein is removed, the tissue of a damaged pancreas can repair itself again. To investigate this further, Zhang et al. – including some of the researchers involved in the 2012 work – created genetically modified mice in which the faulty Kras protein could be experimentally activated or deactivated. The results showed that when Kras was activated, the myeloid cells helped the transformed acinar cells to develop into cancer cells. When Kras was inactivated, myeloid cells helped to repair the damaged tissue. Moreover, myeloid cells used similar molecular signals to either activate the tissue repair or to stimulate the cells to turn into cancer cells.

At the moment, pancreatic cancer cannot be cured. A better understanding of how this disease develops may help scientists develop new treatments.
DOI: https://doi.org/10.7554/eLife.27388.002

pancreatic cancer (*Aguirre et al., 2003*; *Hingorani et al., 2003*; *Hingorani et al., 2005*), a process that is accelerated by the introduction of other common mutations, such as mutation or loss of tumor suppressors p53 and Ink4a/ARF (*Aguirre et al., 2003*; *Hingorani et al., 2005*).

In mice, both ductal and acinar cells can serve as the cell of origin for pancreatic cancer (*De La O et al., 2008*; *Guerra et al., 2007*; *Habbe et al., 2008*; *von Figura et al., 2014*). However, acini are more susceptible to transformation, through a process of dedifferentiation known as acinar-ductal metaplasia (ADM) (*Kopp et al., 2012*). ADM is a reversible physiological process that protects the pancreas upon tissue injury, such as acute pancreatitis (*Halbrook et al., 2017*; *Houbracken et al., 2011*; *Strobel et al., 2007*) in part by reducing the production of digestive enzymes. In presence of oncogenic Kras, pancreatitis-induced ADM becomes irreversible and progresses to PanIN lesions (*Carrière et al., 2009*; *Guerra et al., 2007*). We have previously described the iKras* mouse model, which allows inducible and reversible expression of oncogenic Kras upon administration of doxycycline (DOX) (*Collins et al., 2012a*). By inactivating oncogenic Kras at different stages of carcinogenesis, we showed that sustained Kras activity is necessary to maintain ADM as well as PanIN lesions, and inactivation of oncogenic Kras leads to redifferentiation of acinar cells (*Collins et al., 2012a*). Kras-driven de-differentiation of the acinar cell compartment is mediated, at least in part, by activation of the Kras effector pathway MAPK/ERK (*Collins et al., 2014*). Conversely, the acinar cell specific transcription factors PTF1A and BHLHA15, as well as the pancreatic transcription factor PDX1, protect cellular identity and thus counteract transformation (*Krah et al., 2015*; *Roy et al., 2016*; *Shi et al., 2013*).

Epithelial cells within the pancreas exist in the context of a complex microenvironment that rapidly reacts to tissue damage (for review see[*Ying et al., 2016*]). Myeloid cells are an abundant component of the immune infiltrate during the onset of pancreatic carcinogenesis (*Clark et al., 2007*; *Stromnes et al., 2014*). Macrophages and other myeloid subsets are required for PanIN formation (*Liou et al., 2015*; *Zhang et al., 2017*) and might be sufficient to induce ADM (*Liou et al., 2015*; *Liou et al., 2013*). Macrophages are similarly important for pancreas regeneration following acute damage, such as experimental loss of acinar cells (*Criscimanna et al., 2014*).

While mechanisms regulating acinar cell identity and the regulation of the microenvironment have been addressed separately, we have little understanding of how the cross-talk between different cell types affects these aspects of pancreatic biology. Here, we have set out to study the interactions

between pancreatic epithelial cells and infiltrating myeloid cells and determine the effect of oncogenic Kras expression in modulating this interaction.

## Results

### Myeloid cells are required for PanIN maintenance and progression

To investigate the cross-talk between epithelial cells and myeloid cells, we generated iKras*;CD11b-DTR mice (*Figure 1A*). CD11b-DTR mice express the simian *Diphtheria Toxin Receptor* gene in myeloid cells thus allowing depletion of these cells at will by administration of Diphtheria Toxin (DT) (*Duffield et al., 2005*). To validate myeloid cell depletion in the pancreas, we treated mice with a single dose of Diphtheria Toxin , and the induced acute pancreatitis, a process accompanied by myeloid cell infiltration (*Figure 1—figure supplement 1A*). Compared to control, DT injection resulted in a 40–45% decrease of pancreas infiltrating CD11b$^+$ cells; we observed similar depletion of macrophages and Myeloid-derived suppressor cells (MDSCs), but little change in the dendritic cell population (*Figure 1—figure supplement 1B*). We then depleted myeloid cells in oncogenic Kras-expressing pancreata, following formation of low-grade PanINs. In brief, doxycycline was added to the drinking water to induce oncogenic Kras* expression in adult mice. Acute pancreatitis was induced 72 hr later by caerulein administration for two consecutive days to promote PanIN formation as previously described (*Collins et al., 2012a*). A subset of the mice was sacrificed 3 weeks later, while the remaining animals were administered DT and harvested either 3 days or 1 week later (*Figure 1B*, n = 5–7 mice/cohort). Histopathological analysis 3 weeks post caerulein revealed low-grade PanINs and ADM surrounded by fibrotic stroma throughout the pancreas parenchyma both in iKras* and in iKras*-CD11b mice (*Figure 1C*). DT treatment had no effect on lesion progression in iKras* mice, compared to untreated control. Pancreata from iKras*-CD11b mice harvested 3 days following DT treatment were histologically indistinguishable from control. In contrast, 1 week following myeloid depletion, we observed occasional acini, increased ADM and fewer mucinous lesions and PanINs than in corresponding iKras* tissues (*Figure 1C*, quantification in *Figure 1D*). Furthermore, upon myeloid cell depletion, we observed a reduction in MAPK activation in epithelial cells (as determined by p-ERK1/2 immunostaining) notwithstanding the continuous presence of oncogenic Kras (*Figure 1E*). This reduction in MAPK signaling correlated with an increase of acinar differentiation in the tissue, as determined by staining for Basic helix-loop-helix family member a15 (BHLHA15, also known as MIST1) (*Figure 1F*) and for Amylase, a digestive enzyme (*Figure 1G*). We also observed co-expression of acinar markers (BHLHA15 and Amylase) with the ductal marker CK19, possibly indicating ongoing re-differentiation of acinar cells (*Figure 1F and G*). To distinguish between re-differentiation and outgrowth of cells that had escaped recombination, we stained the tissue for EGFP. The *Rosa26* locus in iKras* mice expresses *rtTa-IRES-EGFP* following Cre recombination (*Collins et al., 2012a*), thus EGFP expression serves as lineage tracing for cells that have undergone recombination and activated oncogenic *Kras* in a rtTa-dependent manner. Our results showed that both PanIN/ADM lesions and recovered acinar cells expressed EGFP, thus validating the redifferentiation of acini from low-grade lesions (*Figure 1—figure supplement 1C*). We also observed a reduction in intracellular mucin, as measured by Periodic Acid–Schiff (PAS) staining (*Figure 2A*). We did not observe changes in apoptosis (Cleaved Caspase three staining, *Figure 2B*). Immunostaining for the macrophage marker F4/80 confirmed depletion of this cell population in the pancreas (*Figure 2C*).

In parallel with changes in the epithelial compartments, myeloid cell depletion led to changes in the stroma. Although tissue fibrosis was still evident by histology (*Figure 1C*), we observed reduced expression of Smooth Muscle Actin (SMA), a fibroblast activation marker (*Figure 1—figure supplement 1D* and quantification in *Figure 1—figure supplement 1E*). Consistently, the expression of genes for the production of extracellular matrix components, such as *Fibronectin 1 (Fn1)*, *Collagen type I alpha one chain (Col1a1)* and *Collagen type III alpha one chain (Col3a1)*, was reduced (*Figure 2D*). *Sonic hedgehog (Shh)*, secreted by pancreatic neoplastic cells to activate surrounding fibroblasts (*Bailey et al., 2008*; *Yauch et al., 2008*), was similarly reduced upon myeloid cell depletion. Even in presence of oncogenic Kras, ligand-mediated activation of EGFR is required to maintain elevated Kras/MAPK activity (*Ardito et al., 2012*). Given the reduction in MAPK signaling levels, we investigated the expression of EGFR ligands by qRT-PCR. Intriguingly, the expression of the EGFR

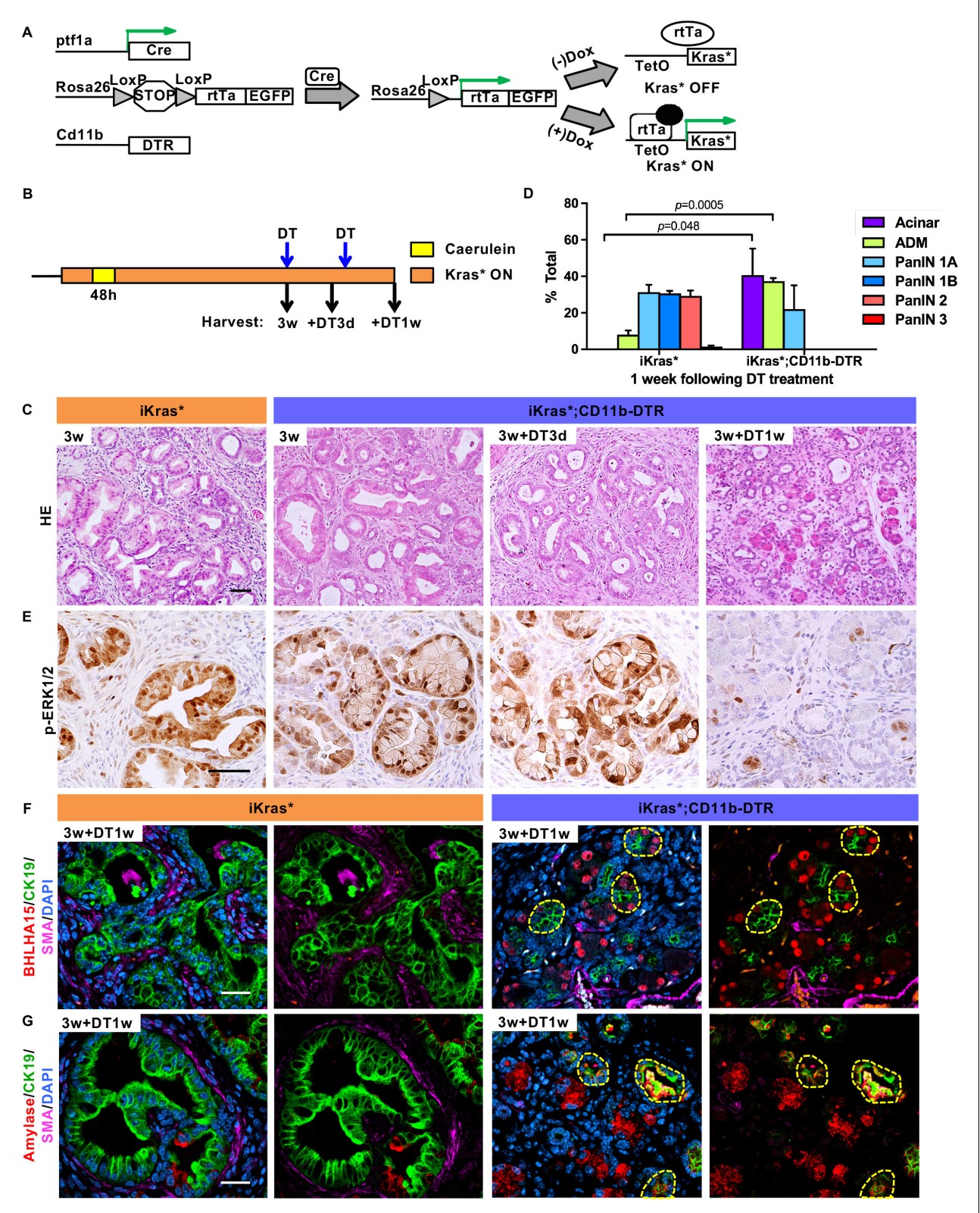

**Figure 1.** Myeloid cells are required for PanIN maintenance. (**A**) Genetic makeup of the iKras*;CD11b-DTR mouse model. (**B**) Experimental design, n = 7 mice/cohort. (**C**) H&E staining of iKras* and iKras*;CD11b-DTR pancreata 3 weeks post pancreatitis induction and iKras*;CD11b-DTR pancreata followed by DT treatment for 3 days and 1 week. Scale bar 50 μm. (**D**) Pathological analysis for iKras* and iKras*;CD11b-DTR pancreata 1 week following DT treatment. Data represent mean ± SEM, n = 3 mice/cohort. The statistical difference between iKras* and iKras*;CD11b-DTR mice per
*Figure 1 continued on next page*

*Figure 1 continued*
lesion type was determined by Two-tailed unpaired *t*-tests. (E) Immunohistochemistry for p-ERK1/2 of iKras* and iKras*;CD11b-DTR pancreata. Scale bar 50 µm. (F) Co-immunofluorescent staining for BHLHA15 (red), CK19 (green), SMA (magenta) and DAPI (blue); (G) Co-immunofluorescent staining for Amylase (red), CK19 (green), SMA (magenta) and DAPI (blue) in iKras* and iKras*;CD11b-DTR pancreata 3 weeks post pancreatitis and followed by DT treatment for 1 week. Scale bar 25 µm. Yellow dashed lines indicate epithelial cell clusters co-expressing BHLHA15 and CK19 (F) or co-expressing Amylase and CK19 (G).
DOI: https://doi.org/10.7554/eLife.27388.003
The following source data and figure supplement are available for figure 1:

**Source data 1.** Histopathological analysis (related to *Figure 1D*).
DOI: https://doi.org/10.7554/eLife.27388.005
**Figure supplement 1.** Depletion of multiple myeloid subsets results in redifferentiation of epithelial cells and remodeling of the stroma.
DOI: https://doi.org/10.7554/eLife.27388.004

ligand *Heparin-Binding epidermal-growth-factor (EGF)-like growth factor* (*Hbegf*) —previously shown to promote pancreatic carcinogenesis (*Ardito et al., 2012*; *Ray et al., 2014*)— decreased upon myeloid cell depletion, suggesting that myeloid cells might be a source of this factor or regulate its expression in other compartments (*Figure 2D*). We observed a similar pattern for *Epiregulin* (*Ereg*), which decreased upon myeloid cell depletion. In contrast, there was no change in three other EGFR ligand genes, *Amphiregulin* (*Areg*), *Transforming growth factor α* (*Tgfα*) and *Egf*. Immunostaining for the active, phosphorylated form of EGFR (p-EGFR), showed expression in in epithelial cells in the control as well as up to three days following myeloid cell depletion, but virtually complete loss of expression by one week (*Figure 2E*). Our data support the notion that myeloid cells – either directly or through interaction with other cell types – are required for activation of EGFR/MAPK signaling in epithelial cells, thus promoting carcinogenesis while preventing acinar re-differentiation and tissue repair.

In advanced malignancy, myeloid cells promote tumorigenesis by inhibiting CD8$^+$ T cell mediated immune responses (*Mitchem et al., 2013*; *Stromnes et al., 2014*; *Zhang et al., 2017*; *Zhu et al., 2014*), and myeloid cell depletion causes CD8$^+$ T cell mediated epithelial cell death. To determine whether a similar immune suppressive mechanism was at play in early lesions, we depleted CD8$^+$ T cells along with myeloid cells in mice bearing low-grade lesions (*Figure 2—figure supplement 1A*). CD8$^+$ T cell depletion alone had no effect on PanIN progression in iKras* mice. Conversely, limited acinar cell recovery was induced by myeloid cell depletion (as described above) and similarly observed when both CD8$^+$ T cell and myeloid cells were depleted in iKras*;CD11b-DTR mice (*Figure 2—figure supplement 1B*). Thus, during the early stages of carcinogenesis, suppression of T cell-mediated immune responses does not appear to be the main function of infiltrating m cells.

## Oncogenic Kras expression in epithelial cells regulates myeloid cell infiltration and polarization

To further investigate the cross-talk between oncogenic Kras expressing epithelial cells and infiltrating myeloid cells, we inactivated oncogenic Kras in PanIN bearing iKras* mice (*Figure 3A*, n = 4–7 mice/cohort) (*Collins et al., 2012a*), and harvested pancreata after 3 days, one week or two weeks. We detected abundant macrophages (CD11b$^+$CD64$^+$F4/80$^+$) in pancreata expressing oncogenic Kras as well as 3 days following Kras inactivation, as determined by immunostaining and flow cytometry (*Figure 3B–C*). The total number of macrophages was significantly lower 1 week following Kras inactivation. We then used a combination of surface markers to measure different subsets of macrophages. In the presence of oncogenic Kras, most infiltrating macrophages were CD11b$^+$CD64$^+$F4/80$^+$CD11c$^+$CD206$^-$, consistent with surface characteristics of tumor associated macrophages (TAMs) (*Franklin et al., 2014*;*Noy and Pollard, 2014*; *Pollard, 2004*). TAM infiltration decreased following Kras* inactivation, while CD11b$^+$CD64$^+$F4/80$^+$CD206$^+$CD11c$^-$ macrophages transiently increased (*Figure 3D*). We sorted total myeloid cells (DAPI$^-$EGFP$^-$CD45$^+$CD11b$^+$) from iKras* pancreata prior to or after Kras inactivation. In myeloid cells extracted from oncogenic Kras expressing pancreata, we detected elevated expression of *Arginase 1* (*Arg1*) and *Chitinase 3-like 3* (*Chil3*) –also known as *Ym1*–mediators of the immune response and commonly expressed in TAMs (*Geiger et al., 2016*; *Munder et al., 1998*; *Raes et al., 2002*); both markers were downregulated in myeloid cells sorted following Kras inactivation (*Figure 3E*). Thus, macrophage polarization is regulated by the Kras

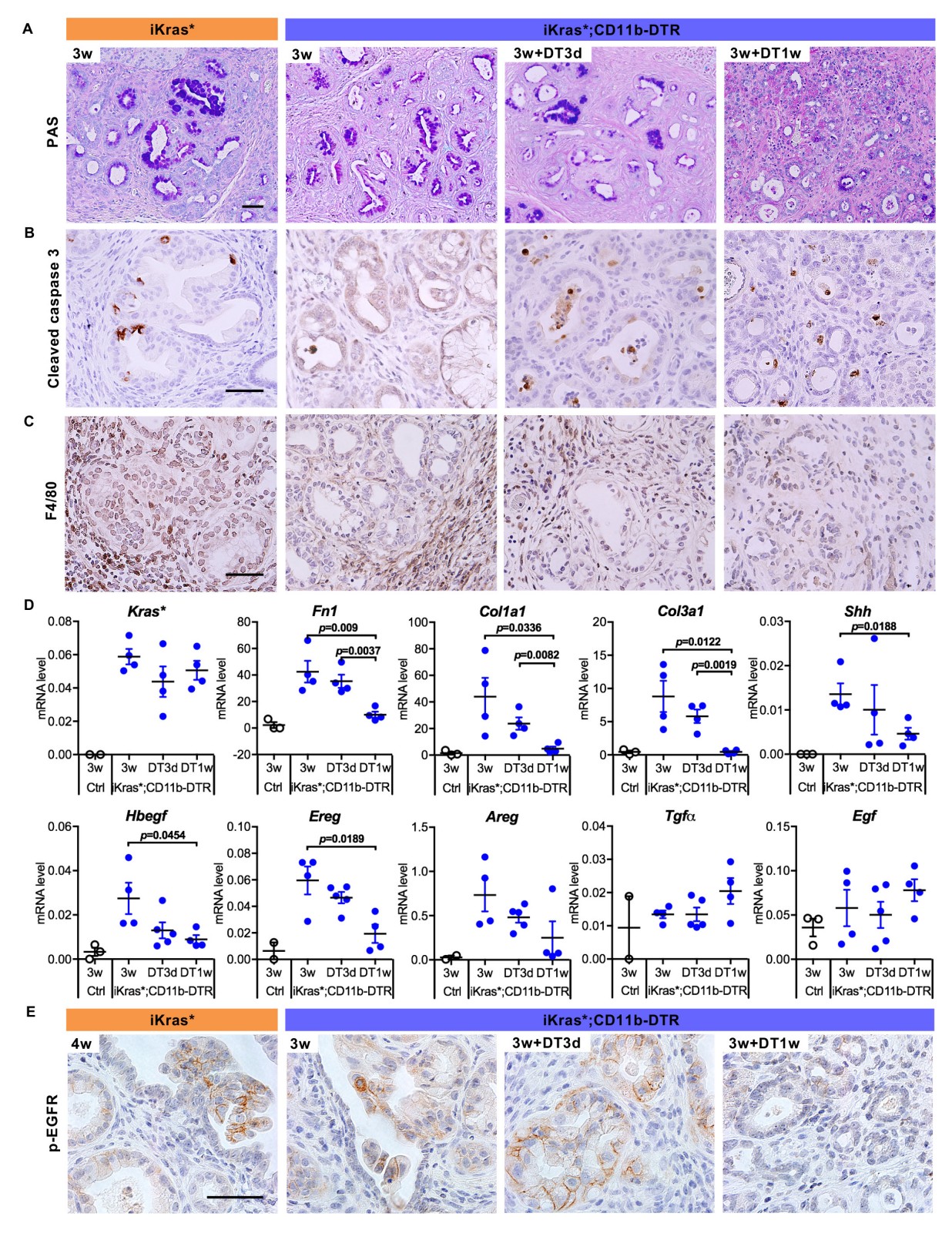

**Figure 2.** Myeloid cell depletion induces PanIN regression and stroma inactivation. (A) Periodic acid–Schiff (PAS) staining; (B) immunohistochemistry staining for Cleaved caspase three and (C) F4/80 in iKras* and iKras*;CD11b-DTR pancreata 3 week post pancreatitis induction and in iKras*;CD11b-DTR pancreata following DT treatment for 3 days and 1 week. Scale bar 50 µm. (D) qRT-PCR for transgenic *Kras*, *Fn1*, *Col1a1*, *Col3a1*, *Shh*, *Hbegf*, *Ereg*, *Areg*, *Tgfα* and *Egf* in littermate control and iKras*;CD11b-DTR pancreata 3 weeks post pancreatitis and in iKras*;CD11b-DTR pancreata following

*Figure 2 continued on next page*

*Figure 2 continued*

DT treatment for 3 days and 1 week. Data represent mean ± SEM, each point indicates one animal. The statistical difference was determined by Two-tailed unpaired *t*-tests. (E) Immunohistochemistry staining for p-EGFR in iKras* and iKras*;CD11b-DTR pancreata post pancreatitis induction and iKras*;CD11b-DTR pancreata following DT treatment for 3 days and 1 week. Scale bar 50 μm.

DOI: https://doi.org/10.7554/eLife.27388.006

The following figure supplement is available for figure 2:

**Figure supplement 1.** CD8 T cell depletion does not affect progression of low-grade PanIN lesions, nor PanIN regression upon myeloid cell depletion.

DOI: https://doi.org/10.7554/eLife.27388.007

status of epithelial cells. To determine whether direct interactions between epithelial cells and myeloid cells mediated expression of *Arg1*, we used an in vitro indirect co-culture system. iKras* primary cells (*Collins et al., 2012*) were cultured in presence or absence of DOX to modulate the expression of oncogenic Kras. Conditioned medium from these cells was then used to culture the mouse macrophage cell line RAW264.7. Analysis of the RNA derived from the macrophages by qRT-PCR revealed that cancer cell conditioned media induced *Arg1* expression in macrophages in an oncogenic Kras dependent manner (*Figure 3F*).

Further characterization of myeloid cells extracted from oncogenic Kras expressing pancreata revealed high levels of *Hbegf*, *Tgfβ* and *Tumor necrosis factor-α (Tnfα)* (*Figure 3G*). The expression of these ligands was reduced in myeloid cells isolated following Kras inactivation, while other secreted molecules, such as *EGF* and *Tgfα*, did not change (*Figure 3E and G*). In parallel with these changes in myeloid cells, we observed a change in the receptor subsets expressed in sorted EGFP⁺ epithelial cells. While *Egfr* expression was high when oncogenic Kras was expressed, and decreased upon its inactivation, *Erbb4* was expressed at a lower level when Kras was active, but increased upon Kras inactivation (*Figure 3—figure supplement 1*).

Thus, oncogenic Kras expression regulates the specific EGFR receptor expressed in the epithelium, as well as regulating polarization and expression of EGFR ligands in infiltrating myeloid cells through a non-cell autonomous mechanism.

## Myeloid cells are required for pancreatic acinar cell re-differentiation

During the early stages of carcinogenesis, oncogenic Kras, through activation of MAPK signaling, promotes dedifferentiation of acinar cells to ADM (*Collins et al., 2014*; *Halbrook et al., 2017*; *Houbracken et al., 2011*; *Strobel et al., 2007*). Conversely, acinar re-differentiation occurs upon inactivation of oncogenic Kras (*Collins et al., 2012a*). While intrinsic factors are known to regulate acinar redifferentiation (*Krah et al., 2015*; *Roy et al., 2016*; *Shi et al., 2013*), the role of the microenvironment is less clear. We investigated the functional role of myeloid cells upon Kras inactivation, during the re-differentiation of acinar cells. We activated oncogenic Kras in adult iKras* or iKras*;CD11b-DTR mice and induced acute pancreatitis, to induce widespread PanIN formation. Three weeks later, we inactivated oncogenic Kras by withdrawing DOX and simultaneously treated the mice with DT, and harvested the pancreata 3 days or 1 week later (*see scheme in Figure 4A*). In DT-treated iKras* mice, inactivation of oncogenic Kras during the early neoplastic stages leads to re-differentiation of acinar cells and remodeling of the extracellular matrix and fibro-inflammatory stroma (*Figure 4B*, *Top Row*). In contrast, depletion of myeloid cells severely impaired this process. A week after Kras inactivation, DT-treated iKras*;CD11b-DTR pancreata remained fibrotic with very few acinar units identified by histology (*Figure 4B*, *Bottom Row*, Trichrome staining in *Figure 4C* and pathological quantification in *Figure 4D*). Furthermore, PAS staining, indicating mucinous ducts and low-grade PanINs, were still present in a subset of the ductal structures (*Figure 4—figure supplement 1A*). Immunostaining for EGFP, a lineage tracing marker for Cre recombination, showed that the epithelial cells in both iKras* and iKras*;CD11b-DTR mice after Kras* inactivation were derived from PanIN cells that had previously expressed the Kras* transgene (*Figure 4—figure supplement 1B*).

In both iKras* and iKras*;CD11b-DTR mice the ductal marker CK19 was prevalent with Kras* ON. Upon Kras inactivation, iKras* mice presented with a transient phase of co-expression of CK19 and amylase, prior to re-establishment of normal pancreas architecture (*Figure 5A*, *Top Row*). In contrast, in iKras*;CD11b-DTR mice, ductal structures with co-expression of CK19 and amylase persisted a week after Kras inactivation (*Figure 5A*, *Bottom Row*). The newly recovered acinar cells in iKras*

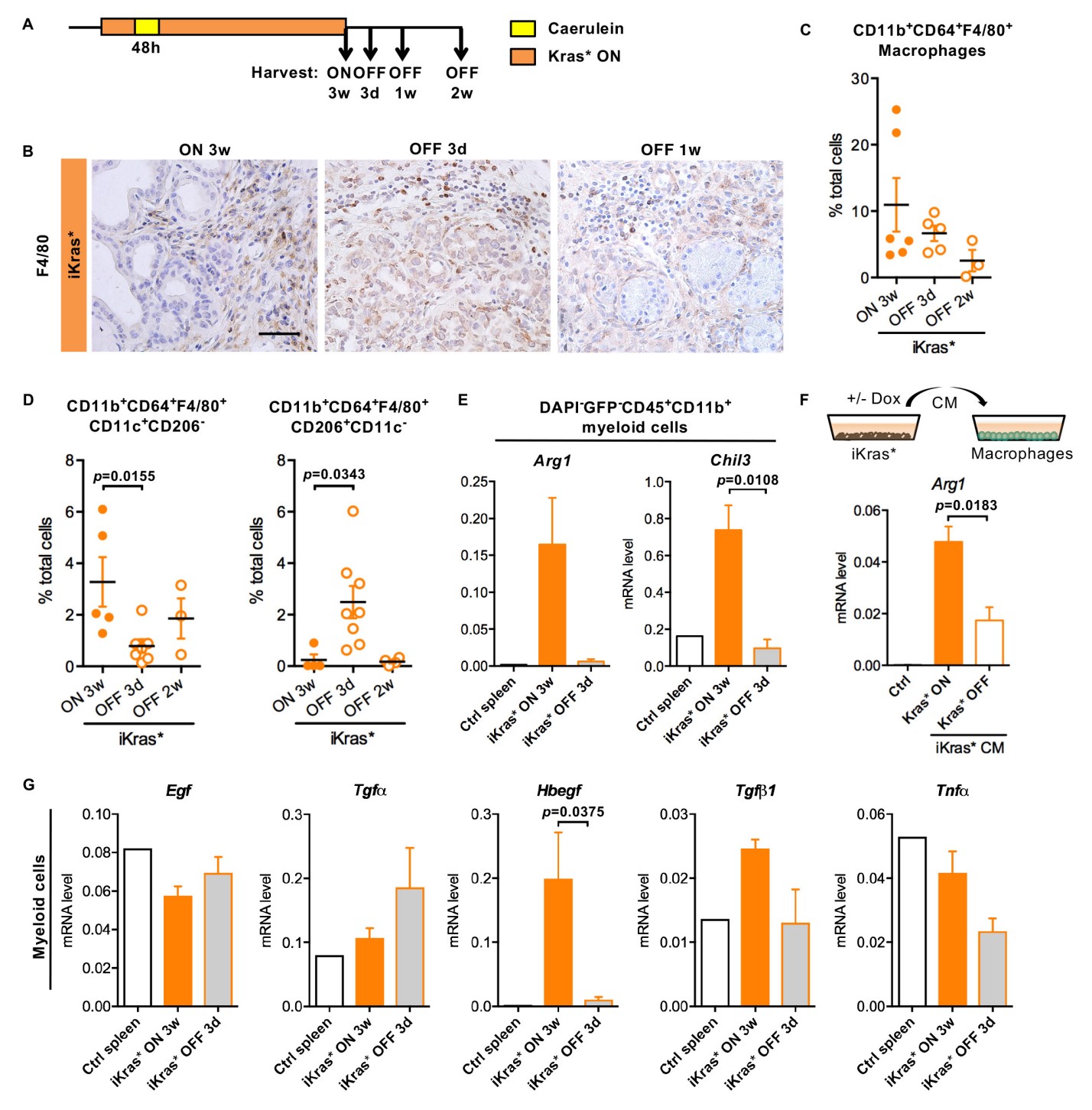

**Figure 3.** Oncogenic Kras in epithelial cells regulates macrophages polarization and function. (**A**) Experimental design, n = 4 ~ 8 mice/cohort. (**B**) Immunohistochemistry staining for F4/80 in iKras* pancreata 3 weeks post pancreatitis and 3 days, 1 week upon Kras* inactivation. Scale bar 50 µm. (**C**) Pancreatic macrophage (CD11b⁺CD64⁺F4/80⁺) infiltration in iKras* pancreata 3 week post pancreatitis induction and 3 days, 2 weeks followed by Kras* inactivation were measured by flow cytometry. (**D**) Percentage of classically activated (CD11b⁺CD64⁺F4/80⁺CD11c⁺CD206⁻) macrophages and alternatively activated (CD11b⁺CD64⁺F4/80⁺CD206⁺CD11c⁻) macrophages in iKras* pancreata 3 weeks post pancreatitis and 3 days, 2 week upon Kras* inactivation were analyzed by flow cytometry. Data represent mean ± SEM, each point indicates one animal. The statistical difference was determined by Two-tailed unpaired $t$-tests. p<0.05. (**E**) qRT-PCR for *Arg1* and *Chil3* expression in myeloid cells flow-sorted from control spleen, iKras* pancreata 3 weeks post pancreatitis and 3 days post Kras* inactivation. Data represent mean ± SEM. The statistical difference was determined by Two-tailed unpaired $t$-tests. (**F**) qRT-PCR for *Arg1* expression in macrophages cultured with control IMDM media or iKras* cancer cell conditioned media (CM) in

*Figure 3 continued on next page*

*Figure 3 continued*

presence or absence of DOX. Data represent mean ± SEM. The statistical difference was determined by Two-tailed unpaired *t*-tests. (G) qRT-PCR for *Egf*, *Tgfα*, *Hbegf*, *Tgfβ1* and *Tnfα* expression in myeloid cells sorted from control spleen, iKras* pancreata 3 weeks post pancreatitis and 3 days post Kras* inactivation. Data represent mean ± SEM. The statistical difference was determined by Two-tailed unpaired *t*-tests.

DOI: https://doi.org/10.7554/eLife.27388.008

The following figure supplement is available for figure 3:

**Figure supplement 1.** Changes in expression of EGFR family receptors in epithelial cells following Kras inactivation.

DOI: https://doi.org/10.7554/eLife.27388.009

mice were highly proliferative (*Figure 5—figure supplement 1A*). In contrast, proliferation was low in iKras*;CD11b-DTR mice a week after Kras* inactivation. Conversely, apoptotic cells were rare in iKras* pancreata, but abundant in the epithelial compartment of iKras*;CD11b-DTR pancreata as shown by E-cadherin and cleaved caspase three co-immunostaining (*Figure 5B*). To determine whether myeloid cell depletion during tissue repair resulted in CD8[+] T cell mediated immune responses against epithelial cells, we depleted CD8[+] T cells along with myeloid cells upon Kras* inactivation (*Figure 5—figure supplement 2A*). In iKras* mice, CD8[+] T cell depletion had no effects on either stroma remodeling or epithelial cell survival, and tissue remodeling occurred as expected. Interestingly, CD8[+] T cell depletion didn't improve cell survival in iKras*;CD11b-DTR mice (*Figure 5—figure supplement 2B*).

Thus, acinar cell plasticity and survival was regulated by infiltrating myeloid cells, independently from their ability to regulate anti-tumor immune responses.

## Pancreatic remodeling requires stromal activation of EGFR/MAPK signaling

Inactivation of oncogenic Kras in pre-neoplastic iKras* pancreata results in a reduction of epithelial p-ERK expression (*Collins et al., 2012a*). Surprisingly, p-ERK downregulation in the epithelium coincided with a transient activation of p-ERK in the stroma (*Figure 5C*). To positively identify the stromal components expressing p-ERK, we performed a panel of co-immunostaining. SMA, a marker of activated fibroblasts, was rapidly reduced upon oncogenic Kras inactivation (*Collins et al., 2012a*); yet we observed expression of p-ERK in a subset of SMA[+] cells (*Figure 5C*, yellow arrows). We observed extensive co-localization of p-ERK with the fibroblast markers Vimentin and Platelet-derived growth factor receptor β (PDGFRβ) (*Figure 5—figure supplement 1B*). In contrast, F4/80[+] macrophages only rarely expressed measurable but low levels of p-ERK (*Figure 5—figure supplement 1B*). Interestingly, myeloid cell ablation prevented p-ERK up-regulation in the stroma upon Kras inactivation (*Figure 5C*). These data are consistent with the hypothesis that myeloid cells provide essential factors that activate EGFR/MAPK signaling in stromal fibroblasts.

We then examined the expression of EGFR ligands and downstream matrix metalloproteinases (MMPs) in our models using qRT-PCR. *Egf* and *Tgfα* levels were significantly up-regulated in iKras* pancreata 3 days following Kras* inactivation, whereas the levels of other EGFR ligands *Hbegf*, *Areg* and *Ereg* were high in neoplastic pancreata (Kras* ON) and dramatically down-regulated when Kras* was inactivated. Depletion of myeloid cells prevented the increase in *Egf* and *Tgfα* upon Kras inactivation (*Figure 6A*). Among the MMPs we examined, *Mmp1* was upregulated in iKras* pancreata 3 days following Kras* inactivation. We observed a similar trend for *Mmp2* and *Mmp9* while the expression of *Mmp12* and *Mmp14* did not change. However, their expression was slightly (but not significantly) decreased upon myeloid cell depletion (*Figure 6A*). To identify the source of EGFR ligands and MMPs during Kras inactivation induced tissue repair, we flow sorted myeloid cells (DAPI[-]EGFP[-]CD45[+]CD11b[+]), fibroblasts (DAPI[-]EGFP[-]CD45[-]CD3[-]CD11b[-]CD31[-]) and epithelial cells (EGFP[+]CD45[-]) for RNA extraction. qRT-PCR analysis showed that *Egf* and *Tgfα* were present in both myeloid cells and fibroblasts, at similar levels independently from the oncogenic Kras status (*Figure 3G* and *Figure 6B*). *Hbegf*, as mentioned earlier, was expressed in a Kras-dependent manner in myeloid cells (*Figure 3G*), but not expressed in fibroblasts (*data not shown*). *EGFR* was expressed in fibroblasts independently of epithelial Kras status (*Figure 6B*). The EGFR/MAPK pathway regulates expression of ECM degrading enzymes in various types of cells including fibroblasts (*Kajanne et al., 2007*). Accordingly, MMPs expression was detected in both myeloid cells and

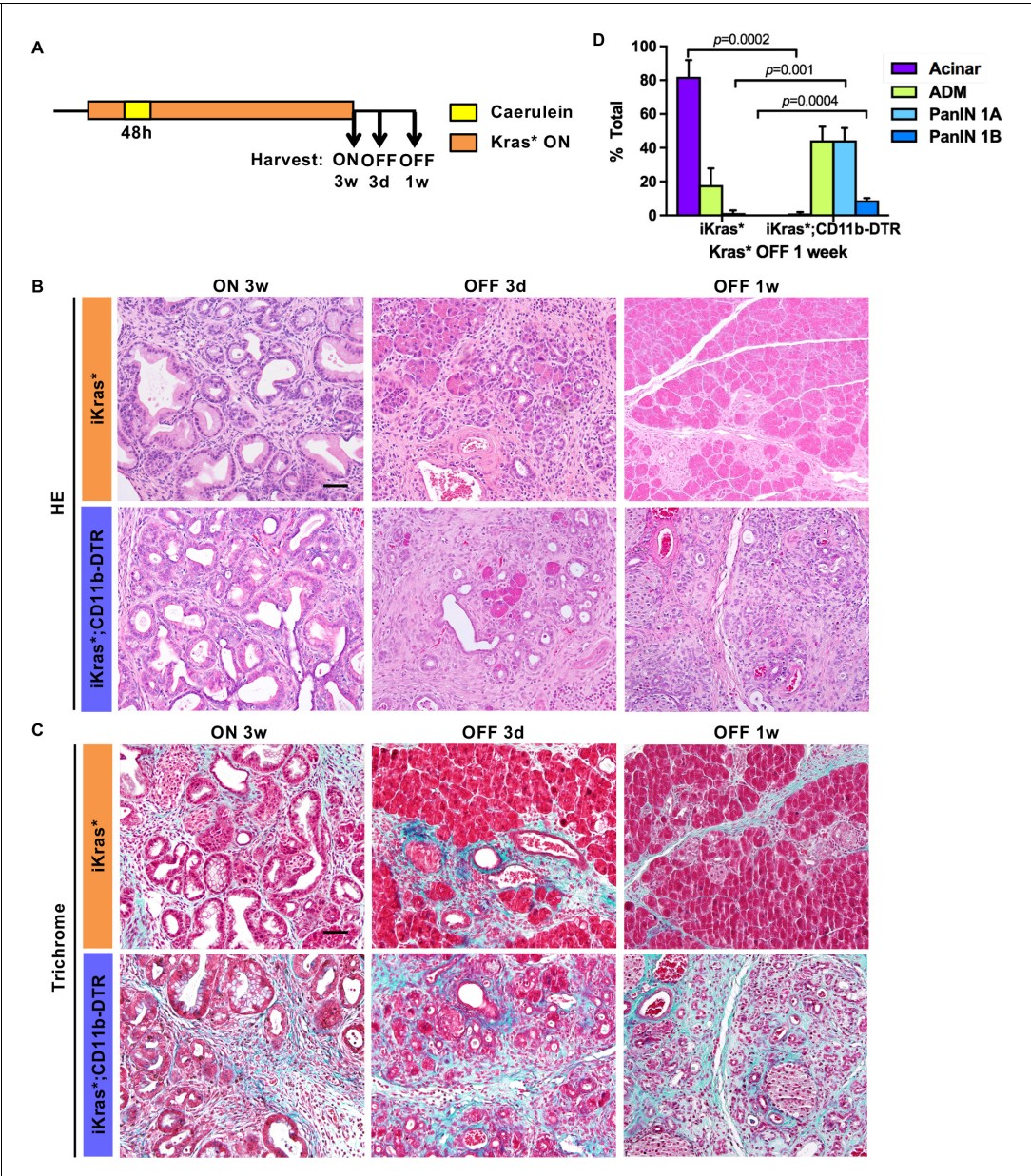

**Figure 4.** Myeloid cells are required for tissue remodeling upon Kras* inactivation. (**A**) Experimental design (n = 4 ~ 8 mice/cohort). (**B**) H&E staining; (**C**) Gomori Trichrome staining in iKras* and iKras*;CD11b-DTR pancreata 3 weeks post pancreatitis induction and 3 days, 1 week following Kras* inactivation and DT treatment. Scale bar 50µm. (**D**) Pathological analysis for iKras* and iKras*;CD11b-DTR pancreata 1 week following Kras* inactivation and DT treatment. Data represent mean ± SEM, n = 4 mice/cohort. The statistical difference between iKras* and iKras*;CD11b-DTR mice per lesion type was determined by Two-tailed unpaired *t*-tests.

DOI: https://doi.org/10.7554/eLife.27388.010

The following figure supplement is available for figure 4:

**Figure supplement 1.** Myeloid cells are required for pancreatic acinar cell re-differentiation upon Kras* inactivation.
DOI: https://doi.org/10.7554/eLife.27388.011

fibroblasts. In particular, *Mmp2* expression in fibroblasts derived from iKras* was up-regulated when Kras* was OFF for 3 days and significantly higher compared to that in fibroblasts derived from iKras*;CD11b-DTR. Further, *Mmp9* expression in fibroblasts decreased upon myeloid cell depletion (***Figure 6B***). Western-blot analysis of the pancreata showed a reduction in overall *Mmp2* protein levels, and specifically the active form of the protein, upon myeloid cell depletion (***Figure 6C***).

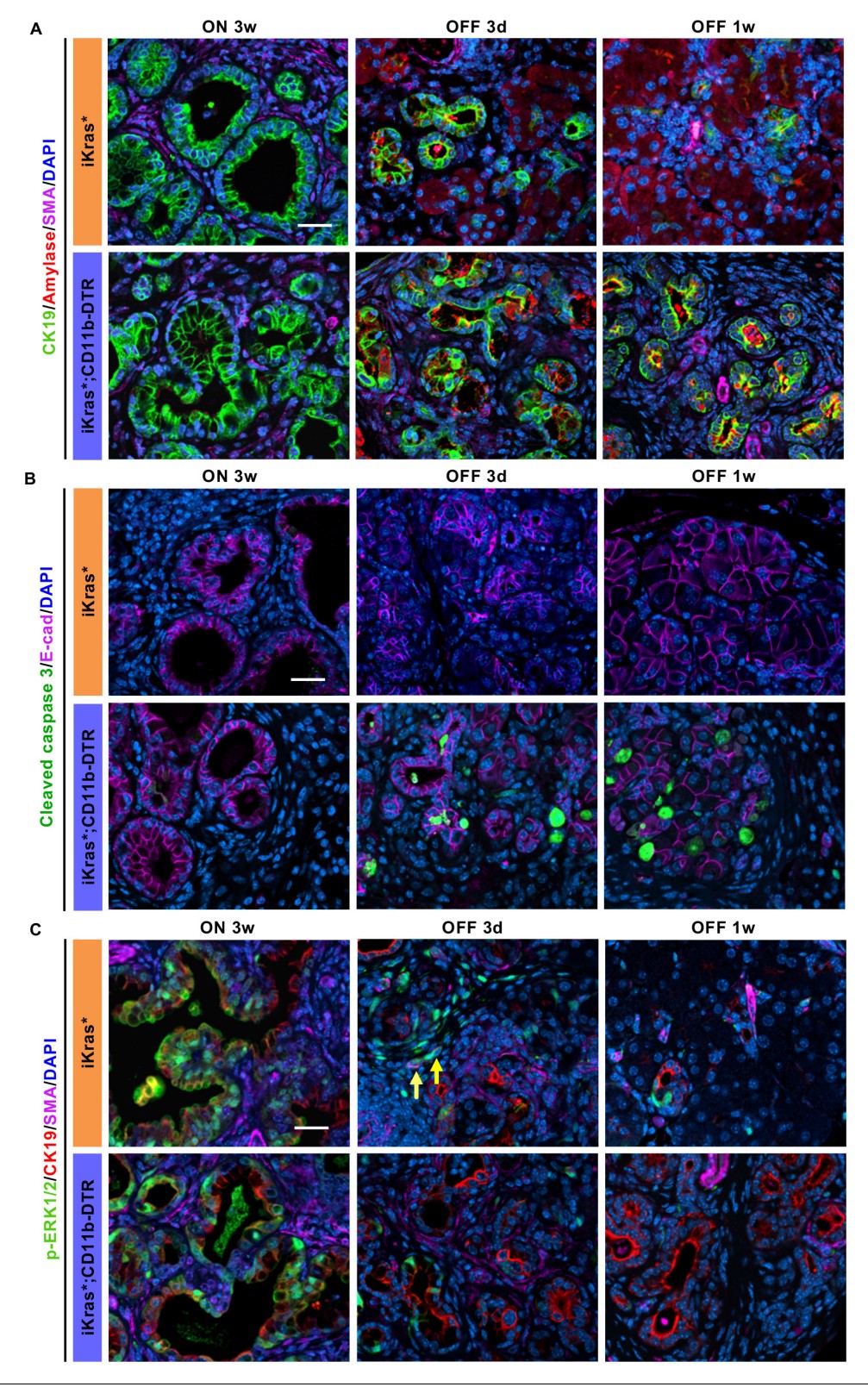

**Figure 5.** Myeloid cells regulate acinar re-differentiation, cell survival and stromal activation of EGFR/MAPK signaling. (**A**) Co-immunofluorescent staining for CK19 (green), Amylase (red), SMA (magenta) and DAPI (blue); (**B**) co-immunofluorescent staining for Cleaved caspase 3 (green), E-cadherin (magenta) and DAPI (blue); (**C**) co-immunofluorescent staining for p-ERK1/2 (green), CK19 (red), SMA (magenta) and DAPI (blue) in iKras* and iKras*;

*Figure 5 continued on next page*

*Figure 5 continued*

CD11b-DTR pancreata 3 weeks post pancreatitis and 3 days, 1 week upon Kras* inactivation and DT treatment. Scale bar 25 µm. Arrows indicate cells co-expressing p-ERK1/2 and SMA.

DOI: https://doi.org/10.7554/eLife.27388.012

The following figure supplements are available for figure 5:

**Figure supplement 1.** Acinar cell regeneration upon Kras* inactivation requires stromal activation of MAPK signaling.

DOI: https://doi.org/10.7554/eLife.27388.013

**Figure supplement 2.** Cell death induced by myeloid cell depletion upon Kras* inactivation is not mediated by CD8[+]T cells.

DOI: https://doi.org/10.7554/eLife.27388.014

Interestingly, Western-blot analysis also revealed a decrease in Collagen I levels following Kras* inactivation in iKras* mice but not in iKras*;CD11b-DTR mice, indicating impaired remodeling in the latter (*Figure 6C*). In addition to myeloid cells, epithelial cells might constitute a source of EGFR ligands. By q-PCR analysis, we detected expression of *Egf*, *Tgfα* and *Hbegf* mRNA in sorted epithelial cells; the expression of *Egf* was decreased upon myeloid cell depletion while expression of the other ligands was unchanged (*Figure 6—figure supplement 1*).

Based on our data, myeloid cells might contribute to EGFR ligand levels both by expressing them directly, and by inducing their expression in other cells types (epithelial, and possibly others).

To determine whether EGFR/MEK activation in fibroblasts was required for tissue remodeling, we inhibited EGFR or MEK – a key component of MAPK signaling- with Erlotinib and Tramatinib, respectively. First, we let iKras* mice develop low-grade PanINs, as described above. Then, upon inactivation of oncogenic Kras, we treated the animals with the inhibitors or vehicle controls (*Figures 7A* and *8A*). EGFR inhibition (EGFRi) blocked MAPK activation in the stroma as measured by reduced p-ERK levels (*Figure 7B*). Similar to myeloid cell depletion, EGFRi treatment resulted in delayed tissue repair. Abundant stroma was still present at 1 week post Kras* inactivation (*Figure 7B*, HE and Trichrome staining). Further, *Mmp2* and *Mmp9* expression was inhibited in EGFRi treated pancreata 3–7 days post Kras* inactivation (*Figure 7C*). However, acinar re-differentiation was not affected by EGFRi treatment, as shown by co-immunofluorescent staining for CK19 and amylase (*Figure 7B*).

We made similar observations upon MEK inhibition (MEKi) upon Kras* inactivation, with reduced ECM degradation and MMPs expression (*Figure 8B and C*), but unimpaired acinar re-differentiation. Therefore, EGFR-MAPK signaling is required for ECM degradation and remodeling. Conversely, in the epithelium, repression of EGFR/MAPK promoted re-differentiation, consistent with previous studies (*Ardito et al., 2012*; *Collins et al., 2014*).

## Discussion

The pancreas is formed by a limited number of progenitor cells and, in the adult, it has a limited ability to regenerate following injury (*Dor et al., 2004*), although it can grow in response to increase need for its exocrine or endocrine function (*Holtz et al., 2014*; *Karnik et al., 2007*). The pancreas is highly plastic; in particular, acinar cells can de-differentiate into duct-like cells during a process known as acinar-ductal metaplasia (ADM). While ADM is important during tissue damage such as acute pancreatitis -where it might protect acinar cells from further damage and set the stage for repair- it also leads to a cell type that is susceptible to transformation by oncogenic Kras (for review see [*Morris et al., 2010*; *Roy and Hebrok, 2015*]). Thus, the mechanisms regulating pancreas plasticity are relevant to both damage/repair in this organ and carcinogenesis.

ADM is characterized by loss of acinar differentiation and acquisition of a duct-like phenotype which is accompanied by expression of pancreas progenitor markers (*Puri and Hebrok, 2010*; *Roy and Hebrok, 2015*; *Stanger and Hebrok, 2013*; *Storz, 2017*). Transcription factors driving acinar differentiation are down-regulated during ADM. BHLHA15 expression is lost during ADM and re-established when ADM re-differentiates to acini. Importantly, BHLHA15 plays a functional role in this process, and while BHLHA15 loss facilitates ADM (and, consequently, carcinogenesis), forced expression of BHLHA15 is protective against both ADM and carcinogenesis (*Shi et al., 2013*). The transcription factor Ptf1a is expressed throughout the pancreatic bud early in development, but it is

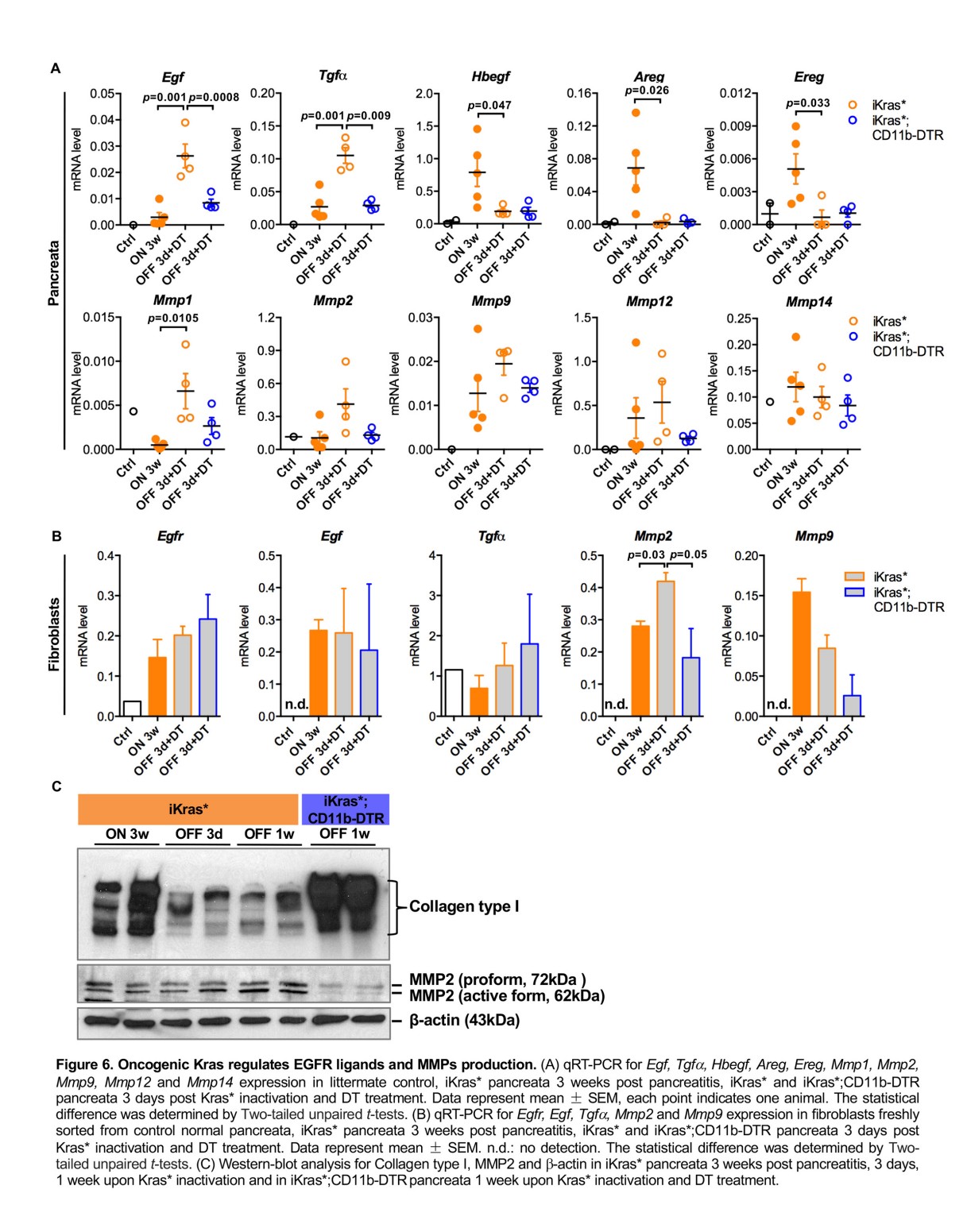

**Figure 6. Oncogenic Kras regulates EGFR ligands and MMPs production.** (A) qRT-PCR for *Egf, Tgfα, Hbegf, Areg, Ereg, Mmp1, Mmp2, Mmp9, Mmp12* and *Mmp14* expression in littermate control, iKras* pancreata 3 weeks post pancreatitis, iKras* and iKras*;CD11b-DTR pancreata 3 days post Kras* inactivation and DT treatment. Data represent mean ± SEM, each point indicates one animal. The statistical difference was determined by Two-tailed unpaired *t*-tests. (B) qRT-PCR for *Egfr, Egf, Tgfα, Mmp2* and *Mmp9* expression in fibroblasts freshly sorted from control normal pancreata, iKras* pancreata 3 weeks post pancreatitis, iKras* and iKras*;CD11b-DTR pancreata 3 days post Kras* inactivation and DT treatment. Data represent mean ± SEM. n.d.: no detection. The statistical difference was determined by Two-tailed unpaired *t*-tests. (C) Western-blot analysis for Collagen type I, MMP2 and β-actin in iKras* pancreata 3 weeks post pancreatitis, 3 days, 1 week upon Kras* inactivation and in iKras*;CD11b-DTR pancreata 1 week upon Kras* inactivation and DT treatment.

**Figure 6.** Oncogenic Kras regulates EGFR ligands and MMPs production. (**A**) qRT-PCR for *Egf, Tgfα, Hbegf, Areg, Ereg, Mmp1, Mmp2, Mmp9, Mmp12* and *Mmp14* expression in littermate control, iKras* pancreata 3 weeks post pancreatitis, iKras* and iKras*;CD11b-DTR pancreata 3 days post Kras* inactivation and DT treatment. Data represent mean ± SEM, each point indicates one animal. The statistical difference was determined by Two-tailed unpaired *t*-tests. (**B**) qRT-PCR for *Egfr, Egf, Tgfα, Mmp2* and *Mmp9* expression in fibroblasts freshly sorted from control normal pancreata, iKras*

*Figure 6 continued*

pancreata 3 weeks post pancreatitis, iKras* and iKras*;CD11b-DTR pancreata 3 days post Kras* inactivation and DT treatment. Data represent mean ±SEM. n.d.: no detection. The statistical difference was determined by Two-tailed unpaired *t*-tests. (**C**) Western-blot analysis for Collagen type I, MMP2 and β-actin in iKras* pancreata 3 weeks post pancreatitis, 3 days, 1 week upon Kras* inactivation and in iKras*;CD11b-DTR pancreata 1 week upon Kras* inactivation and DT treatment.

DOI: https://doi.org/10.7554/eLife.27388.015

The following figure supplement is available for figure 6:

**Figure supplement 1.** Changes in expression of EGFR ligands in epithelial cells following Kras inactivation.

DOI: https://doi.org/10.7554/eLife.27388.016

restricted to acinar cells in the adult organ (*Kawaguchi et al., 2002*). Ptf1a loss facilitates ADM and carcinogenesis (*Krah et al., 2015*). Further, PDX1, a key determinant of pancreas development expressed at low levels in adult acini, is similarly important to maintain acinar cell identity (*Roy et al., 2016*). Thus, signals intrinsic to epithelial cells regulate the differentiation status of acinar cells. We have, and others, have previously shown that oncogenic Kras induces ADM through activation of MAPK signaling (*Ardito et al., 2012*; *Collins et al., 2014*; *Collisson et al., 2012*) and consequent repression of acinar-specific transcription factors. Conversely, inhibition of MAPK signaling using MEK inhibitors allows re-expression of acinar-specific transcription factors and re-differentiation of acinar cells (*Collins et al., 2014*). Thus, a complex network of intrinsic signals regulates acinar differentiation in the adult pancreas.

In our initial characterization of the iKras* mouse model, we investigated the role of oncogenic Kras during very early stages of carcinogenesis. While oncogenic Kras promotes transdifferentiation of acinar cells to acinar-ductal metaplasia, inactivation of oncogenic Kras in ADM or even low-grade PanIN lesions leads to regression of these lesions and re-differentiation of the epithelial compartment to acinar cells (*Collins et al., 2012a*). Inactivation of oncogenic Kras also results in profound remodeling of the surrounding fibroinflammatory reaction. Here, we set out to understand the interaction between oncogenic Kras expressing epithelial cells and the surrounding microenvironment. We show that reciprocal interactions between oncogenic Kras expressing epithelial cells and the surrounding microenvironment control pancreatic plasticity (see working model in *Figure 9*). *First,* we determined that Kras expressing epithelial cells alter myeloid cell polarization in the pancreas, inducing expression of *Arginase1, Chil3* and *Hbegf*. These markers have been previously described in tumor associated macrophages (TAMs, for review see [*Mantovani et al., 2017*]). *Second,* Inactivation of oncogenic Kras led to the loss of *Arg1*, *Chil3* and *Hbegf from myeloid cells.* Conversely, a subset of macrophages positive for the surface markers CD206 and CD11c, transiently accumulated in the pancreas, coinciding with the remodeling process. Interestingly, the surface marker expression of this population is consistent with M2 macrophages previously shown to be important in regeneration of pancreatic acini and islets following experimental ablation (*Criscimanna et al., 2014*), and similarly involved in tissue repair in other organs (for review, see [*Wynn and Vannella, 2016*]).

In this study, we show that myeloid cells play an instructive role regulating epithelial cell identity and plasticity. In the presence of oncogenic Kras, myeloid cells are required to maintain dedifferentiation of ADM/low-grade PanIN lesions. Depletion of myeloid cells induces BHLHA15 expression and occasionally expression of the digestive enzyme amylase in low-grade PanINs, notwithstanding expression of oncogenic Kras, thus presumably preventing further progression to malignancy. We show that myeloid cells are required for the expression of EGF ligands, and activation of MAPK signaling in pancreatic epithelial cells. This finding fits with the notion that oncogenic Kras is insufficient to induce a high enough level of MAPK activation to induce transformation (*Daniluk et al., 2012*), thus EGFR ligands are required for carcinogenesis (*Ardito et al., 2012*).

In tumor-bearing mice, myeloid cells inhibit CD8$^+$ T cell mediated anti-tumor immune responses, and this function explains their requirement in cancer growth (*Stromnes et al., 2014*; *Zhang et al., 2017*). However, both during the progression of early PanIN lesions and during their regression upon Kras inactivation, myeloid cell-requirement was independent from the presence of CD8$^+$ T cells, indicating that they play a function distinct from immune suppression. Upon Kras inactivation, myeloid cells including re-polarized M2 macrophages are required for tissue remodeling. *First*, we show that myeloid cells are required for epithelial cell re-differentiation and survival. Thus, myeloid cell depletion results in epithelial cell death and persistence of clusters of cells co-expressing acinar

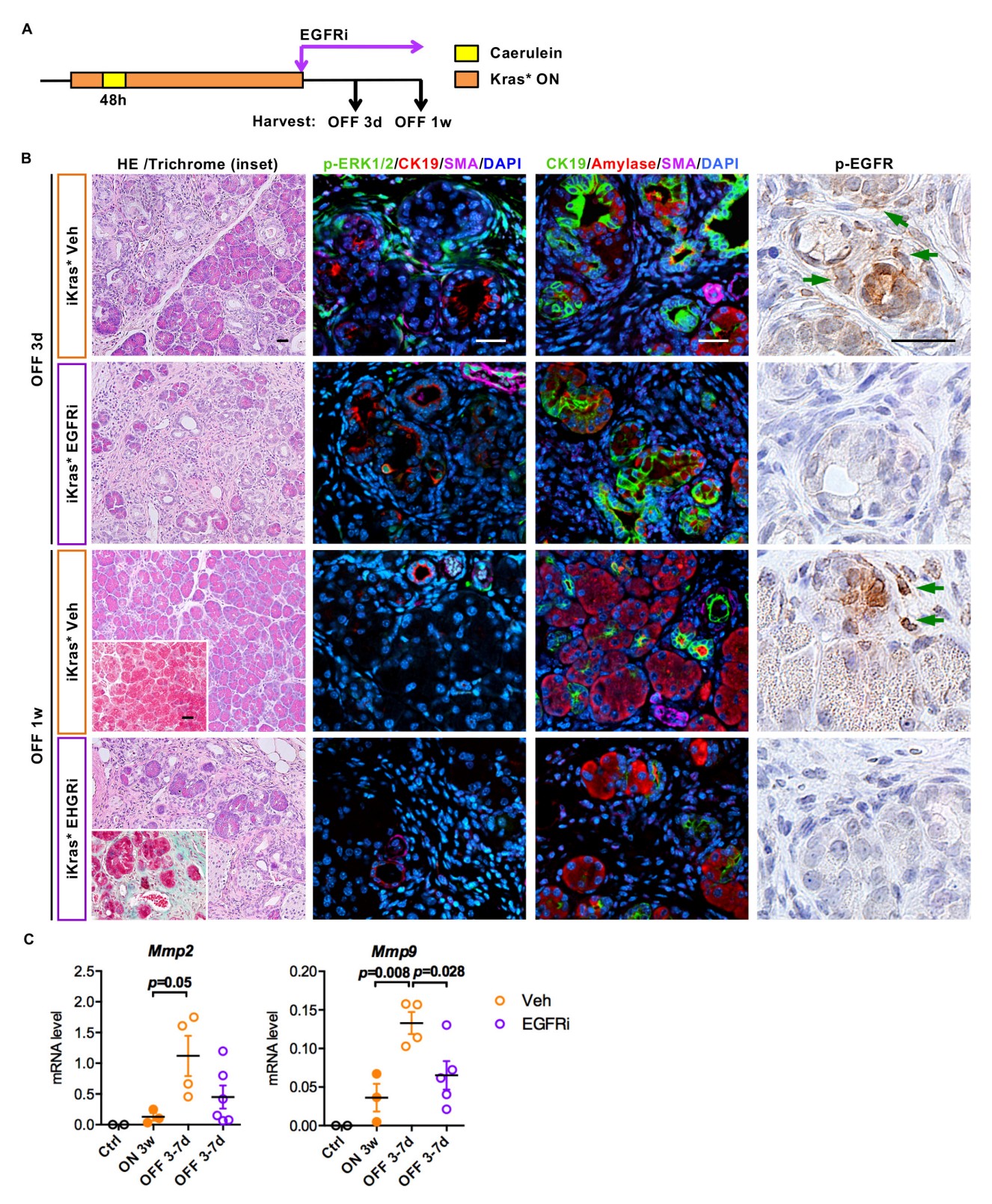

**Figure 7.** EGFR inhibition impairs ECM remodeling and tissue repair upon Kras* inactivation. (A) Experimental design, n = 4 ~ 6 mice/cohort. (B) H&E staining, Gomori Trichrome staining (inset), co-immunofluorescent staining for p-ERK1/2 (green), CK19 (red), SMA (magenta) and DAPI (blue), co-immunofluorescent staining for CK19 (green), Amylase (red), SMA (magenta) and DAPI (blue) and immunohistochemistry staining for p-EGFR in iKras* pancreata 3 weeks post pancreatitis induction followed by Kras* inactivation and EGFR inhibitor (EGFRi) treatment for 3 days and 1 week. Scale bar 25

*Figure 7 continued on next page*

*Figure 7 continued*

μm. (C) qRT-PCR for *Mmp2* and *Mmp9* expression in littermate control pancreata, iKras* 3 weeks post pancreatitis, iKras* 3–7 days followed by Kras* inactivation and treated with either vehicle or EGFRi. Data represent mean ± SEM, each point indicates one animal. The statistical difference was determined by Two-tailed unpaired *t*-tests.

DOI: https://doi.org/10.7554/eLife.27388.017

and ductal markers. *Second,* we show that myeloid cells are required for remodeling of the extracellular matrix. To gain mechanistic insight, we investigated the cross-talk between epithelial cells and myeloid cells. We have previously shown that myeloid cells are required to sustain activation of EGFR/MAPK signaling in epithelial cells (*Zhang et al., 2017*). In turn, MAPK signaling is necessary for PanIN formation and progression (*Ardito et al., 2012*; *Collins et al., 2014*; *Collisson et al., 2012*). Surprisingly, remodeling of the extracellular matrix was also regulated by EGFR/MAPK signaling. Inactivation of oncogenic Kras in the pancreas led to changes of expression of specific EGFR ligands in the pancreas. In presence of active Kras, *Hbegf*, *Areg* and *Ereg* were the main ligands. Upon Kras* inactivation, their expression decreased while that of *Egf* and *Tgfα* increased. In parallel, we observed changes in expression of EGFR family receptors. Remarkably, inactivation of oncogenic Kras resulted in transient activation of MAPK signaling in stromal fibroblasts, simultaneous to loss of activation in the epithelium. Consistent with the notion that MAPK signaling in the stroma is important for remodeling, treatment with Erlotinib (EGFR inhibitor) or Tramatinib (MEK inhibitor) resulted in the persistence of pancreatic fibrosis. The observation that activation of EGFR/MAPK signaling in different cellular compartments might, in turn, favor carcinogenesis or remodeling has potential clinical implications, suggesting that specific inhibition of distinct EGFR ligands or receptors might be preferable to overall inhibition. While our data support the notion that myeloid cells are a source of EGFR ligands, they also support the idea that myeloid cells induce EGFR ligands in other cellular compartments, including epithelial cells; future studies will need to address the role of specific EGFR ligands and their specific cell sources.

In summary, in this study we show that the cross-talk between epithelial cells and myeloid cells regulates pancreatic plasticity and fibrosis. Further, we show that this cross-talk is important for pancreatic tissue repair following injury, but can be co-opted, in presence of oncogenic Kras, to promote carcinogenesis. Manipulating this cross-talk to promote repair while inhibiting carcinogenesis should therefore be prioritized in future studies.

## Materials and methods

### Mice

iKras*;CD11b-DTR mice were generated by crossing iKras* mice (ptf1a-Cre;R26-rtTa-IRES-EGFP; TetO-Kras[G12D]) (*Collins et al., 2012a*) with CD11b-DTR mice (B6.FVB-Tg(ITGAM-DTR/EGFP)34Lan/ J, *Jackson* Laboratory) (*Duffield et al., 2005*). Double mutant littermates of iKras* were used as controls. Male and female mice were included equally. All animal studies were conducted in compliance with the guidelines of Institutional Committees on Use and Care of Animals at the University of Michigan.

### Animal experiments

Acute pancreatitis was induced in 4–6 week-old mice by caerulein injection and Kras[G12D] expression was induced by doxycycline as previously described (*Collins et al., 2012a*). Three weeks post pancreatitis induction doxycycline was withdrawn from the drinking water for tissue repair study. Mice were also treated with EGFR inhibitor Erlotinib (50 mg/kg, oral gavage, daily) (Selleckchem), or MEK inhibitor Tramatinib (GSK1120212) (1 mg /kg, i.p. daily) (Selleckchem) or vehicle. For myeloid cell depletion, CD11b-DTR and iKras*;CD11b-DTR mice were treated with diphtheria toxin (DT) (25 ng/g i.p.) (Enzo Life Science) and repeated every 4 days. For CD8[+] T cell depletion, anti-CD8 mAb (BioXcell #BE0061; clone 2.43; 200 μg/mouse) was injected i.p. twice per week.

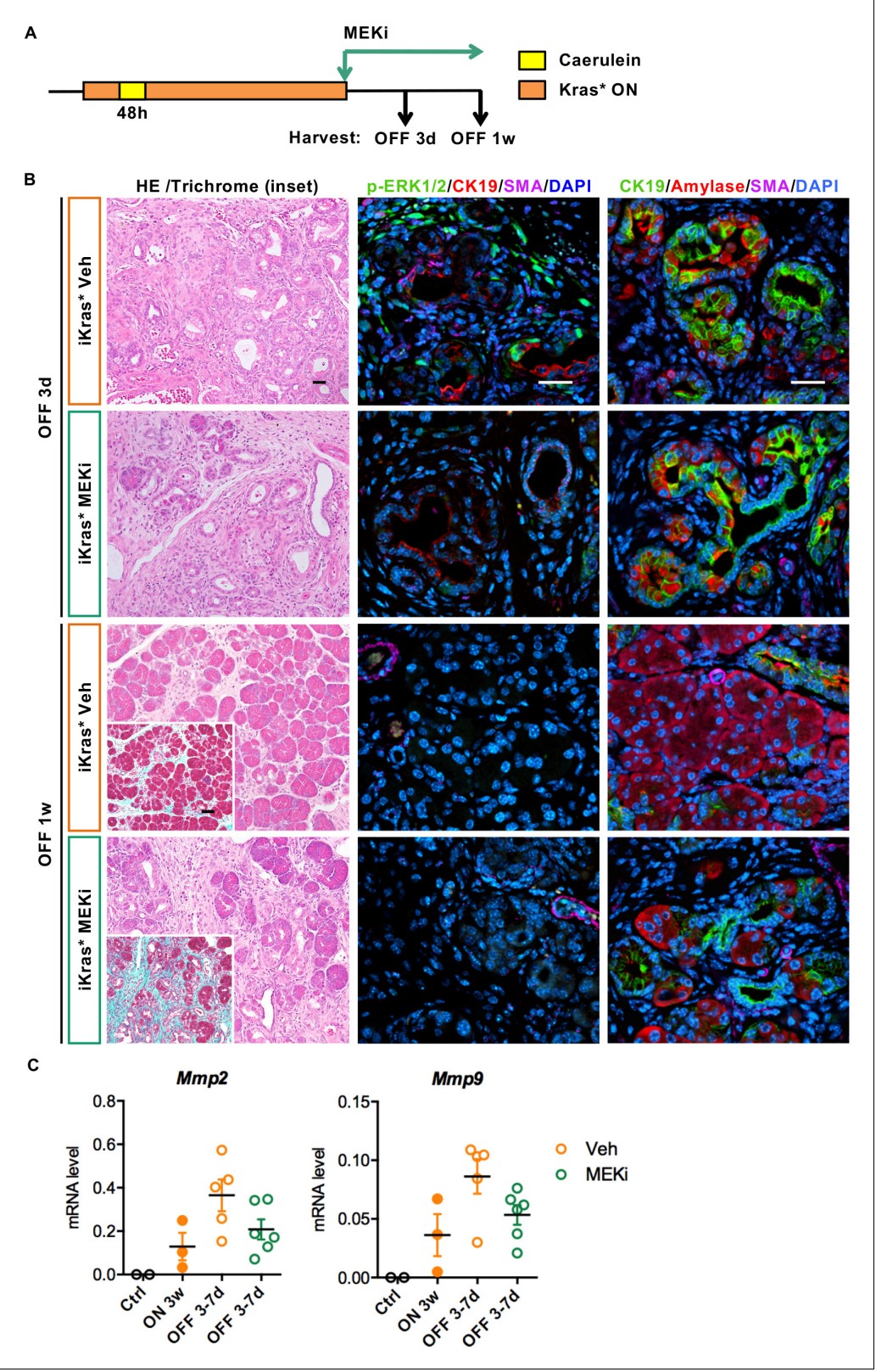

**Figure 8.** Stroma remodeling upon Kras* inactivation depends on MEK/ERK activity in stroma. (**A**) Experimental design, n = 4 ~ 5 mice/cohort. (**B**) H&E staining, Gomori Trichrome staining (inset), co-immunofluorescent staining for p-ERK1/2 (green), CK19 (red), SMA (magenta) and DAPI (blue) and co-immunofluorescent staining for CK19 (green), Amylase (red), SMA (magenta) and DAPI (blue) in iKras* pancreata 3 weeks post pancreatitis induction

*Figure 8 continued on next page*

*Figure 8 continued*
followed by Kras* inactivation and MEK inhibitor (MEKi) treatment for 3 days and 1 week. Scale bar 25 μm. (**C**) qRT-PCR for *Mmp2* and *Mmp9* expression in littermate control pancreata, iKras* 3 weeks post pancreatitis, iKras* 3–7 days followed by Kras* inactivation and treated with either vehicle or MEKi. Data represent mean ± SEM, each point indicates one animal.
DOI: https://doi.org/10.7554/eLife.27388.018

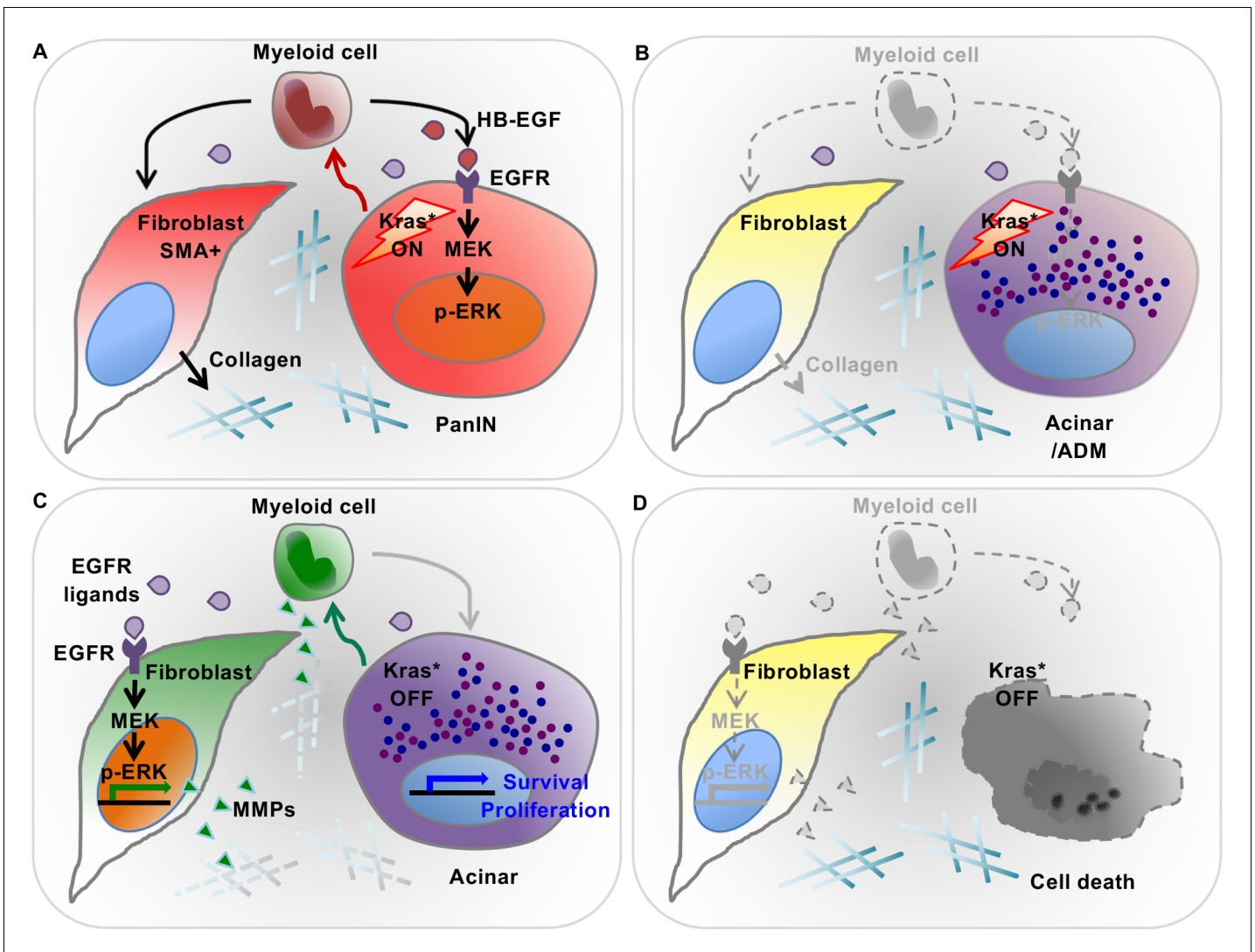

**Figure 9.** Diagram depicting our working model. (**A**) In the presence of epithelial oncogenic Kras expression, myeloid cells are required to maintain high MAPK activity in the epithelial cells themselves, and thus maintain de-differentiation and promote PanIN formation. At the same time, myeloid cells are required for the activation status of stromal fibroblasts. These effects are, at least in part, mediated by secretion of EGFR ligands. Myeloid cells are a source of HB-EGF (maroon drops), which is possibly also secreted by epithelial cells. Further, other EGFR ligands (purple drops) are secreted by cells within the microenvironment. (**B**) Myeloid cell depletion leads to re-differentiation of PanIN cells and loss of fibroblast activation, notwithstanding continuous expression of oncogenic Kras. (**C**) Oncogenic Kras inactivation in epithelium redirects myeloid cell polarization and function; thus inducing MAPK activation in fibroblasts to promote extracellular matrix remodeling. (**D**) Myeloid cell depletion following Kras* inactivation results in impaired pancreatic tissue repair.
DOI: https://doi.org/10.7554/eLife.27388.019

## Cell culture

All cells were cultured in IMDM supplemented with 10% FBS and 1% penicillin/streptomycin (Gibco). Primary mouse pancreatic cancer cell line iKras* derived from iKras*p53* (ptf1a-Cre; TetO-Kras$^{G12D}$; Rosa26$^{rtTa/+}$; p53$^{R172H/+}$) tumor (*Collins et al., 2012*) was used to generate conditioned medium (CM) in presence or absence of Doxycycline at 1 µg/ml (Sigma) for 2–3 days. These cells were used at low passage, genotyped for the Kras, Cre and p53 transgenes, and tested negative for mycoplasma. Mouse macrophage cell line RAW264.7 (ATCC Cat# TIB-71, RRID:CVCL_0493) were similarly used at low passage and mycoplasma negative. CM was filtered through 0.2 µm filter before use. 1–2 $\times$ 10$^5$ cells of RAW264.7 were plated in 6-well plates overnight and then cultured with CM (iKras* CM diluted 1:1 in fresh IMDM with 10% FBS) for 24 hr before harvest for RNA isolation.

## Histopathological analysis

Hematoxylin and eosin (H&E), Periodic Acid Schiff (PAS), Gomori's Trichrome, immunohistochemical and immunofluorescent staining were performed on formalin-fixed, paraffin embedded mouse pancreatic tissues as described before (*Zhang et al., 2013a*). Antibodies used are listed in *Supplementary file 1*. For immunofluorescence, Alexa Fluor (Invitrogen) secondary antibodies were used. Cell nuclei were counterstained with Prolong Gold with DAPI (Invitrogen). Images were taken with Olympus BX-51 microscope, Olympus DP71 digital camera, and DP Controller software. The immunofluorescent images were acquired using the Olympus IX-71 confocal microscope and Fluo-View FV500/IX software. For histopathological analysis, five non-overlapping H&E images (20x objective) per slide were examined by a pathologist (W.Y.) as described before (*Zhang et al., 2013a*). Image-Pro Plus 4.5 was used to measure the percentage of positive area of immunofluorescent staining per high power field image. Three samples per group, and 4–6 images per sample were analyzed.

## Western blotting

Western blotting was conducted as previously described (*Collins et al., 2012a*), and Collagen I was detected under non-reduced and non-denatured condition. Antibody information is included in *Supplementary file 1*.

## Flow cytometric analysis and sorting

Single-cell suspensions of fresh spleen or pancreas were prepared as previously described (*Zhang et al., 2013b*) and stained with fluorescently conjugated antibodies listed in *Supplementary file 1*. Flow cytometric analysis was performed on a Cyan ADP analyzer (Beckman Coulter) and data were analyzed with Summit 4.3 software. Cell sorting was performed using a MoFlo Astrio (Beckman Coulter). Myeloid cells (DAPI$^-$EGFP$^-$CD45$^+$CD11b$^+$), epithelial cells (DAPI$^-$EGFP$^+$CD45$^-$) and fibroblasts (DAPI$^-$EGFP$^-$CD45$^-$CD11b$^-$CD31$^-$CD3$^-$) were sorted and lysed in RLT buffer (Qiagen). Total RNA was prepared using RNeasy (Qiagen) and reverse-transcribed using High Capacity cDNA Reverse Transcription kit (Applied Biosystems).

## Quantitative RT-PCR

Samples for quantitative PCR were prepared with 1X SYBR Green PCR Master Mix (Applied Biosystems) and various primers (primer sequences are listed in *Supplementary file 2*). All primers were optimized for amplification under reaction conditions as follows: 95$^\circ$C 10mins, followed by 40 cycles of 95$^\circ$C 15 secs and 60$^\circ$C 1 min. Melt curve analysis was performed for all samples after the completion of the amplification protocol. Cyclophilin A was used as the housekeeping gene expression control.

## Statistical analysis

Graphpad Prism six software was used for all statistical analysis. All data were presented as means ± standard error (SEM). Intergroup comparisons were performed using Two-tailed unpaired $t$-test, and $p < 0.05$ was considered statistically significant.

## Acknowledgements

We thank Marsha Thomas and Kevin T Kane for lab support. The ptf1a-Cre mouse was generous gift from Dr. Chris Wright (Vanderbilt University). The BHLHA15 antibody was a gift from Dr. Stephen Konieczny (Purdue University), and the CK19 antibody was obtained from the Iowa Developmental Hybridoma Bank.

## Additional information

### Funding

| Funder | Grant reference number | Author |
|---|---|---|
| University of Michigan | Biological Scholar Program | Marina Pasca di Magliano |
| American Cancer Society | Kras and the Inflammatory Microenvironment in Pancreatic Cancer | Marina Pasca di Magliano |
| Elsa U. Pardee Foundation | Targeting Tumor Infiltrating Immune Cells as a Therapeutic Strategy in Pancreatic Cancer | Marina Pasca di Magliano |
| National Cancer Institute | NCI-1R01CA151588 | Marina Pasca di Magliano |
| National Cancer Institute | 3-P30-CA-046592-28-S2 | Marina Pasca di Magliano Howard C Crawford |
| National Institutes of Health | University of Michigan Program in Cellular and Molecular Biology training grant NIH T32 GM007315 | Esha Mathew |
| National Institutes of Health | University of Michigan Gastrointestinal Training Grant NIH T32 DK094775 | Esha Mathew |
| National Cancer Institute | P30-CA-046592 | Marina Pasca di Magliano |

The funders had no role in study design, data collection and interpretation, or the decision to submit the work for publication.

### Author contributions

Yaqing Zhang, Data curation, Formal analysis, Investigation, Writing—original draft; Wei Yan, Formal analysis; Esha Mathew, Kevin T Kane, Arthur Brannon III, Formal analysis, Investigation; Maeva Adoumie, Alekya Vinta, Investigation; Howard C Crawford, Data curation, Funding acquisition, Writing—review and editing; Marina Pasca di Magliano, Conceptualization, Supervision, Funding acquisition, Writing—original draft, Project administration

### Author ORCIDs

Marina Pasca di Magliano http://orcid.org/0000-0001-9632-9035

### Ethics

Animal experimentation: This study was performed in strict accordance with the recommendations in the Guide for the Care and Use of Laboratory Animals of the National Institutes of Health. All of the animals were handled according to approved institutional animal care and use committee (IACUC) protocols (PRO00005959) of the University of Michigan. The protocol was approved by the University Committee on Use and Care of Animals (UCUCA) of the University of Michigan on 11/17/2014. The current protocol is valid until 11/17/2017. For the pancreatitis experiments and tumorigenesis experiments, strict guidelines were followed to minimize suffering of the animals. Animal activity levels and weight were monitored throughout the experiments.

Decision letter and Author response
Decision letter https://doi.org/10.7554/eLife.27388.023
Author response https://doi.org/10.7554/eLife.27388.024

## Additional files

### Supplementary files

• Supplementary file 1. Primary antibodies used in this study
DOI: https://doi.org/10.7554/eLife.27388.020

• Supplementary file 2. Primer sequences for quantitative RT-PCR
DOI: https://doi.org/10.7554/eLife.27388.021

• Transparent reporting form
DOI: https://doi.org/10.7554/eLife.27388.022

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
