## [Decision Letter]

Thank you for submitting your article "Epithelial-Myeloid cell crosstalk regulates acinar cell plasticity and pancreatic remodeling in mice" for consideration by *eLife*. Your article has been favorably evaluated by Didier Stainier (Senior Editor) and three reviewers, one of whom is a member of our Board of Reviewing Editors. The reviewers have opted to remain anonymous.

The reviewers have discussed the reviews with one another and the Reviewing Editor has drafted this decision to help you prepare a revised submission.

Summary:

This is a very interesting and well-executed study demonstrating highly context dependent contrasting roles for myeloid cells, likely macrophages, in initiation and regression of pancreatic cancer development and tissue remodeling. Using their elegant model of pancreatic ductal adenocarcinoma (PDA) with inducible KRASG12D expression coupled with depletion of CD11b+ myeloid cells, the authors interrogate effects on early disease development and tissue remodeling to show that myeloid cells are critical not only to disease development but also for tissue remodeling following silencing of oncogenic Kras. The three key findings are: (1) established PanINs, induced by oncogenic Kras, require macrophages for their maintenance, and undergo at least partial re-differentiation when macrophages are ablated; (2) the regression of PanINs that occurs when oncogenic Kras is turned off *also* requires macrophages; and (3) these contrasting roles reflect EGFR/MAPK signaling, dependent on macrophages, being activated in epithelial vs. stromal (fibroblast) cells.

Essential revisions:

While most of the data are well presented and provide important and novel insights, it would be very useful if the authors were to address concerns that many of the links that they make are descriptive correlations rather than causally established. Some concerns, as identified below, require new data, while other concerns require substantial textual modifications.

Concerns requiring new data:

1) While the manuscript shows that CD8 T cells do not mediate cell killing in the Kras*-off/DT model, it is critical to establish this in the Kras*-on/DT model. It is important to determine if the major role of macrophages at this early stage is to suppress anti-tumor immunity (which the authors' own prior studies have established as quite potent under certain circumstances, even in early PanINs). This is important because the data here are consistent not only with the authors' preferred model of PanINs depending on macrophage-delivered EGFR ligands, but also with a model in which PanIN persistence requires ongoing immune suppression by the macrophages. (See also point 3, below, regarding an alternative model for the pro-remodeling role of macrophages.)

2) In Figure 2, while it is interesting that the authors observe increased levels of the EGFR ligand Hb-egf, there are several ligands that can activate EGFR. To interrogate the interactions more thoroughly, the authors should examine the expression of other ligands. Moreover, as the authors have examined levels of other EGFR ligands in subsequent experiments (Figure 6), addressing the levels of other ligands related to Figure 2 will provide a basis for better comparing the differential role of macrophages in disease development and resolution.

Concerns requiring extensive textual modifications:

3) If one compares the expression of EGFR ligands between whole pancreas (Figure 6), sorted myeloid cells (Figure 3) and sorted fibroblasts (Figure 6), an alternative model can be suggested: both EGF and TGFalpha are elevated in whole pancreas specifically upon Kras* inactivation, in a macrophage-dependent manner, suggesting that *epithelial cells* are a potential source of ligands acting on fibroblasts during repair, and that the role of macrophages is to induce epithelial cells to produce these ligands. This is also consistent with the pEGFR staining in Figure 7, where positive fibroblasts are seen adjacent to positive epithelial cells, suggesting autocrine/paracrine signaling from the epithelium. This is at least a likely model, especially since there are no data provided showing that macrophages activate induce pEGFR in co-cultured pancreatic fibroblasts. The authors should acknowledge and address such alternative models and/or provide additional data to justify their model of a macrophage source for EGFR ligands.

4) Experiments described in this manuscript have only involved depletion of myeloid cells, not specifically macrophages. Statements such as "we show that M2 macrophages are required for epithelial cell re-differentiation and survival" are thus not well supported. Clear distinctions between myeloid cells and macrophages should be made throughout the text. In fact, although it has presumably been defined in other studies, it would be useful for the authors to clearly indicate just what types of myeloid cells are being depleted in DT-injected CD11b-DTR mice. Presumably, dendritic cells or granulocytes are not being depleted, but the term "myeloid cells" is very broad and it is advisable to have some more specific description.

5) The discussion of macrophage subtypes is confusing and contradictory (e.g. subsection “Oncogenic Kras expression in epithelial cells regulates myeloid cell infiltration and polarization” and Discussion, third paragraph). On the one hand, cell surface markers are interpreted as showing a transient increase in "alternatively activated macrophages" after Kras* inactivation. But in the same paragraph, the authors state that Arg1 and Ym1 mRNA expression, "commonly expressed in alternatively activated macrophages," is high in macrophages of Kras* expressing mice, and these markers clearly decline after Kras* inactivation (Figure 3). The data are the data, and probably indicate that our definitions of macrophage subtypes are imprecise, but the authors need to provide some interpretation of these inconsistencies rather than over-use the term "alternatively activated."

6) Related to Figure 1, the authors write "myeloid cell depletion resulted in a striking reduction in MAPK…". This is an observation showing correlation and not causality and should be presented as such.

7) The authors write, "These data suggest that myeloid cells provide essential signals to activate EGFR/MAPK signaling in epithelial cells…" The observations provide correlation and not causality, which should be reflected in the statement – there are no data presented to suggest this is necessarily a specific effect of myeloid secreted HB-Egf on the epithelial cells.

---

## [Author Response]

Essential revisions:While most of the data are well presented and provide important and novel insights, it would be very useful if the authors were to address concerns that many of the links that they make are descriptive correlations rather than causally established. Some concerns, as identified below, require new data, while other concerns require substantial textual modifications.Concerns requiring new data:1) While the manuscript shows that CD8 T cells do not mediate cell killing in the Kras*-off/DT model, it is critical to establish this in the Kras*-on/DT model. It is important to determine if the major role of macrophages at this early stage is to suppress anti-tumor immunity (which the authors' own prior studies have established as quite potent under certain circumstances, even in early PanINs). This is important because the data here are consistent not only with the authors' preferred model of PanINs depending on macrophage-delivered EGFR ligands, but also with a model in which PanIN persistence requires ongoing immune suppression by the macrophages. (See also point 3, below, regarding an alternative model for the pro-remodeling role of macrophages.)

This is an important point, and we agree with the reviewers that it is an essential complement to our initial data. We have now addressed this question experimentally, and believe that the new data strengthens the manuscript.

In brief, iKras* and iKras*;CD11b-DTR mice were placed on DOX to activate oncogenic Kras expression. We then induced acute pancreatitis to promote ADM and PanIN formation. After 3 weeks, when the mice have widespread lesions though the pancreas, we randomized mice of each genotype to either anti-CD8 or isotype control (IgG) and Diphtheria Toxin. In iKras* mice (where the DT treatment has no effect on myeloid cells), CD8^+^ T cell depletion did not change lesion progression, consistent with the notion that limited/no immune suppression of pancreatic cancer occurs in immune competent mice. In iKras*;CD11b-DTR mice, DT treatment resulted in depletion of myeloid cells and 1 week after depletion we observed regression of PanINs and partial repopulation of acini. CD8^+^ T cell depletion did not impair nor promote PanIN regression. Thus, the requirement for myeloid cells during the early stages of both carcinogenesis and repair does not depend on their ability to block anti-tumor T cell responses. The new data is included in Figure 2—figure supplement 1,and addressed in the last paragraph of the subsection “Myeloid cells are required for PanIN maintenance and progression”.

2) In Figure 2, while it is interesting that the authors observe increased levels of the EGFR ligand Hb-egf, there are several ligands that can activate EGFR. To interrogate the interactions more thoroughly, the authors should examine the expression of other ligands. Moreover, as the authors have examined levels of other EGFR ligands in subsequent experiments (Figure 6), addressing the levels of other ligands related to Figure 2 will provide a basis for better comparing the differential role of macrophages in disease development and resolution.

The reviewers are correct, that a more comprehensive analysis of EGFR ligands should have been included. We have now performed and included qPCR analysis for *Ereg, Areg, Tgfα*, and *Egf*. Interestingly, myeloid cell depletion in low-grade PanINs results in a reduction of *Ereg –* similar to what we previously observed for *Hb-egf-* but no change in *Tgfα*, and *Egf.* There was also no significant change in *Areg* although we observed a slight trend towards decrease.The new data is included in Figure 2, and described in the second paragraph of the subsection “Myeloid cells are required for PanIN maintenance and progression”.

Concerns requiring extensive textual modifications:3) If one compares the expression of EGFR ligands between whole pancreas (Figure 6), sorted myeloid cells (Figure 3) and sorted fibroblasts (Figure 6), an alternative model can be suggested: both EGF and TGFalpha are elevated in whole pancreas specifically upon Kras* inactivation, in a macrophage-dependent manner, suggesting that epithelial cells are a potential source of ligands acting on fibroblasts during repair, and that the role of macrophages is to induce epithelial cells to produce these ligands. This is also consistent with the pEGFR staining in Figure 7, where positive fibroblasts are seen adjacent to positive epithelial cells, suggesting autocrine/paracrine signaling from the epithelium. This is at least a likely model, especially since there are no data provided showing that macrophages activate induce pEGFR in co-cultured pancreatic fibroblasts. The authors should acknowledge and address such alternative models and/or provide additional data to justify their model of a macrophage source for EGFR ligands.

We have addressed this comment both by performing additional experiments and by modifying the text. We have analyzed sorted epithelial cells for the expression on EGFR ligands, and detected expression of *Egf, Tgfα* and *HB-Egf*; further, at least for a subset of these ligands, epithelial expression decreased upon myeloid cell depletion. The new data is included in Figure 6—figure supplement 1.

Thus, it is likely that the effect of myeloid cells on EGFR signaling is two-fold: 1) direct secretion of ligands and 2) promotion of ligand expression from other cell types, including epithelial cells. We have modified accordingly the subsection “Pancreatic remodeling requires stromal activation of EGFR/MAPK signaling”, the model in Figure 9, and the description of the model in the Discussion section.

4) Experiments described in this manuscript have only involved depletion of myeloid cells, not specifically macrophages. Statements such as "we show that M2 macrophages are required for epithelial cell re-differentiation and survival" are thus not well supported. Clear distinctions between myeloid cells and macrophages should be made throughout the text. In fact, although it has presumably been defined in other studies, it would be useful for the authors to clearly indicate just what types of myeloid cells are being depleted in DT-injected CD11b-DTR mice. Presumably, dendritic cells or granulocytes are not being depleted, but the term "myeloid cells" is very broad and it is advisable to have some more specific description.

We thank the reviewers for pointing out a potential confusing description of our results. We have checked through the text and amended “macrophage” to myeloid cells wherever appropriate. We use the term “macrophage” only where it can be fully justified, namely when we use macrophage-specific markers such as F4/80. Further, although, as the reviewers suggest, previous characterization of the CD11b-DTR model was performed in the initial description, we have performed experiments to determine the profile of myeloid depletion in our hands. For this purpose, we have used a single-dose DT treatment in control or CD11b-DTR mice following induction of acute pancreatitis. The single dose DT is less efficient than the multiple doses we use in the rest of the manuscript, but here our goal was not to achieve maximum depletion, but to determine whether different cellular subsets are depleted with different efficacy. As shown in Figure 1—figure supplement 1,macrophages and myeloid derived suppressor cells were depleted at about the same rate as total myeloid cells. In contrast, dendritic cells were *not/only slightly* affected. These results are described in the first paragraph of the subsection “Myeloid cells are required for PanIN maintenance and progression”.

5) The discussion of macrophage subtypes is confusing and contradictory (e.g. subsection “Oncogenic Kras expression in epithelial cells regulates myeloid cell infiltration and polarization” and Discussion, third paragraph). On the one hand, cell surface markers are interpreted as showing a transient increase in "alternatively activated macrophages" after Kras* inactivation. But in the same paragraph, the authors state that Arg1 and Ym1 mRNA expression, "commonly expressed in alternatively activated macrophages," is high in macrophages of Kras* expressing mice, and these markers clearly decline after Kras* inactivation (Figure 3). The data are the data, and probably indicate that our definitions of macrophage subtypes are imprecise, but the authors need to provide some interpretation of these inconsistencies rather than over-use the term "alternatively activated."

The reviewers rightly point out that our nomenclature was confusing. In part, this reflects the difficulty to define distinct subtypes of macrophages, as these cells are plastic and exist in a continuum of different forms. To add to the difficulty, multiple subsets of myeloid derived suppressor cells are also present within the tissue. We now have changed the text in the following way:

a) In the Results section, we describe both surface markers and expression of specific functional markers (such as Arg1). We further indicate that expression of Arg1 is common in tumor associated macrophages (TAMs).

b) In the Discussion, we compare the myeloid cells and macrophages subsets to similar subtypes in the literature. We also made sure to distinguish between myeloid cells and macrophages based on the specific experiment.

Accordingly, we amended the text extensively in the subsection “Oncogenic Kras expression in epithelial cells regulates myeloid cell infiltration and polarization” and in the Discussion section. While we trust these changes improve the clarity of our description, we have to acknowledge that defining macrophage subsets is complicated, and we –as well as other groups- are actively working at trying to define their nature, their recruitment and changes over time.

6) Related to Figure 1, the authors write "myeloid cell depletion resulted in a striking reduction in MAPK…". This is an observation showing correlation and not causality and should be presented as such.

The reviewers are correct, and we have modified the text to avoid suggesting causality. The text now reads: “… upon myeloid cell depletion, we observed a reduction in MAPK activation…”.

7) The authors write, "These data suggest that myeloid cells provide essential signals to activate EGFR/MAPK signaling in epithelial cells…" The observations provide correlation and not causality, which should be reflected in the statement – there are no data presented to suggest this is necessarily a specific effect of myeloid secreted HB-Egf on the epithelial cells.

As above, this is an excellent point, and we have amended the text (subsection “Myeloid cells are required for PanIN maintenance and progression”) to avoid confounding correlation with causality.